# Explaining Deep Q-Learning Experience Replay with SHapley Additive exPlanations

Robert S. Sullivan  and Luca Longo *

Artificial Intelligence and Cognitive Load Research Lab, School of Computer Science, Technological University Dublin, Grangegorman, D07 ADY7 Dublin, Ireland; robssully@gmail.com
* Correspondence: luca.longo@tudublin.ie

**Abstract:** Reinforcement Learning (RL) has shown promise in optimizing complex control and decision-making processes but Deep Reinforcement Learning (DRL) lacks interpretability, limiting its adoption in regulated sectors like manufacturing, finance, and healthcare. Difficulties arise from DRL's opaque decision-making, hindering efficiency and resource use, this issue is amplified with every advancement. While many seek to move from Experience Replay to A3C, the latter demands more resources. Despite efforts to improve Experience Replay selection strategies, there is a tendency to keep the capacity high. We investigate training a Deep Convolutional Q-learning agent across 20 Atari games intentionally reducing Experience Replay capacity from $1 \times 10^6$ to $5 \times 10^2$. We find that a reduction from $1 \times 10^4$ to $5 \times 10^3$ doesn't significantly affect rewards, offering a practical path to resource-efficient DRL. To illuminate agent decisions and align them with game mechanics, we employ a novel method: visualizing Experience Replay via Deep SHAP Explainer. This approach fosters comprehension and transparent, interpretable explanations, though any capacity reduction must be cautious to avoid overfitting. Our study demonstrates the feasibility of reducing Experience Replay and advocates for transparent, interpretable decision explanations using the Deep SHAP Explainer to promote enhancing resource efficiency in Experience Replay.

**Keywords:** deep reinforcement learning; experience replay; SHapley Additive exPlanations; eXplainable artificial intelligence



## 1. Introduction

In recent years, Reinforcement Learning (RL) has emerged as a powerful technique for optimizing complex control and decision-making processes [1]. However, this potential has not been fully realized in regulated industries such as manufacturing [2], finance [3], and healthcare [4] due to a critical challenge - the lack of explainability in Deep Reinforcement Learning (DRL). The absence of transparency in DRL models has resulted in difficulties in debugging and interpreting the decision-making process. These challenges, highlighted by [5,6] have led to the development of inefficient models that strain available resources. Moreover, as Deep Learning continues to advance, particularly with improvements to critical components like Experience Replay, the challenges related to debugging, interpretation, and inefficiency become even more pronounced [7]. To address these pressing issues, an emerging field known as Explainable Reinforcement Learning (XRL) aims to provide solutions. One prominent tool in the XRL toolkit is SHapley Additive exPlanations (SHAP), which offers insights into the contributions of input features to a model [8].

In this paper, we leverage Deep SHAP, developed by [9,10], as an additional tool to visualize Experience Replay, enhancing the debugging and interpretability of Deep Q-Learning (DQL). While some previous works extensively compared various aspects of DQL, such as [11] comparing different algorithms, Ref. [12,13] comparing different selection strategies, and [14] who explored both increasing capacity and sampling ratio from $1 \times 10^6$ and 0.25 respectively, our approach in this paper is distinct. We focus solely on testing the

effect of reducing Experience Replay capacity on the average cumulative reward received, Both this and visualizing experience replay have not been explored before. We also chose to use a Deep Convolutional Q-learning (DCQL) algorithm as [14] did but to only test it in 20 Atari games due to hardware constraints. Our aim is to find an optimal point from $1 \times 10^6$ used by [11] or from $1 \times 10^4$ recommended by [12], where the average cumulative game score is maximized, and Experience Replay capacity $M$ is minimized.

The rest of this paper is structured as follows: Section 2 presents a brief review of related works on Deep Q-learning, Experience Replay, and Explainable Reinforcement Learning. Section 3 outlines the experimental design where Experience Replay is reduced and its effect on rewards received is measured. Section 4 focuses on implementing a Deep Convolutional Q-Learning, and xAI Explainer model in 20 Atari games, through an experiment that generates reward scores and explanations. This section then discusses findings in the context of xAI, and Section 5 presents the conclusion.

## 2. Related Works

### 2.1. Classic Reinforcement Learning

In classical Reinforcement Learning (RL), the model or learning system called an Agent, learns without a labeled dataset to complete tasks through trial and error. This is possible by receiving a reward indicating how well it is performing. Reward is received from the environment where the Agent resides when it performs an action given a situation or state [15]. Solving problems or controlling a process can be structured as either a Multi-Arm Bandit (MAB), Contextual Bandit, or a tuple environment called a Markov Decision Making Process (MDP) [15–17]. Classically, these environment structures are tabular or grid-world based. MDP was considered as it has became a popular choice due to high-dimensional contexts and complex sequential decision-making problems being easier to deal with. Firstly, MDP contains states $s$ such as a vector of encoded values or an image that contains all the information needed to describe what happened at time $t$. Secondly, it holds a set of actions $a$ chosen at time $t$. Thirdly, it contains a Transition Rule or probability distribution, describing the probability of the next state $s_{t+1}$ at time $t + 1$ given the current state and action selected. Finally, it holds a reward $r$ that returns a score for selecting an action in a given state. MDP assumes that the stochastic process to be modeled has the Markov property, meaning the conditional probability distribution of being in a future state does not depend on any previous states or actions, except the current ones [18]. Many real-world applications can apply MDP according to [19], such as insurance, sales, inventory management, and patient admissions. The following defines what the Transition Rule and Reward Function (Equations (1) and (2)) are:

$$T : (a_t \in A, s_t \in S, s_{(t+1)} \in S) \rightarrow P(s_{(t+1)}|s_t, a_t) \tag{1}$$

$$R : (a_t \in A, s_t \in S) \rightarrow r_t \in R \tag{2}$$

An Agent in an MDP environment selects actions given certain states based on a Policy $\pi$ (Equation (3)):

$$\pi : s_t \in S \rightarrow a_t \in A \tag{3}$$

A policy is a set of instructions to follow, a strategy or just a function that maps states to actions. It can comprise of either random actions, selected by a subject matter expert or be dynamically selected by the Agent. The best Policy $\pi^*$ (Equation (4)) out of a collection of policies $\Pi$, is one that maximizes the cumulative reward [16]. When debugging and interpreting the results of a reinforcement learning system maximizing reward is the desired outcome of training a model.

$$\pi^* = \underset{\pi \in \Pi}{\operatorname{argmax}} \sum_{t \geq 0} R(\pi(s_t), s_t) \tag{4}$$

The reward for each state in Reinforcement Learning is predetermined when defining the MDP (Equation (5)). Gamma $\gamma$ is used to signify the discount factor and is always set to less than 1 (e.g., 0.95 or 0.99 considered in this paper). When $\gamma$ is close to 1 the Agent will optimize for future rewards but anything greater than 1 will cause the discount reward to reach infinity. When $\gamma$ is close to 0 the Agent will optimize its current reward [16].

$$R_t \ = \ r_t + \gamma r_{t+1} + \gamma^2 r_{t+2} ... + \gamma^{n-t} r_n \tag{5}$$

To maximize its reward, the Agent explores the environment to receive more information and make better decisions or to exploit good decisions it already knows. Several algorithms exist to develop exploration strategies in Classical Reinforcement Learning, The simplest exploration strategies are A/B/n testing, Epsilon-greedy, Upper Confidence bounds, and Thomson Sampling [15]. Others include Dynamic Programming (DP) Policy Iteration and Value Iteration algorithms but they require multiple iterations over the entire state space to find optimal policies. Asynchronous Dynamic Programming addresses this by focusing on more likely encountered states, but not all states are equally important. Additionally, knowing the transition probabilities can be challenging in most cases making these solutions inappropriate for complex environments. Monte Carlo methods solve this by allowing an optimal policy to be found solely from experience with the environment [15] but it lacks sample efficiency. Model-based methods, such as derivative-based and derivative-free methods, offer better sample efficiency, in some cases and can deal with model uncertainty, especially with limited data using neural networks such as Bayesian Neural Networks [20] and Ensemble models [21]. Model-based methods are good when exploration is costly, as in controlling physical systems [13], but they are more computationally expensive [15]. In contrast, model-free algorithms offer sample efficiency since they don't assume knowledge of the environment's transition dynamics and only learn from sampled experiences. On-Policy and Off-Policy methods differ in their approach to exploration and exploitation during training, with On-Policy methods being more suitable for situations with high exploration costs or continuous action spaces and Off-Policy methods being more efficient in reusing past experiences [15].

This paper considered a popular model-free, off-policy-based algorithm called Q-Learning, which is grounded in Temporal Difference learning. The quality of taking an action given a state in an environment is called Q-values $Q(a, s)$ and are stored, prior to deep learning, in the cells of a Q-table for each state-action pair. These Q-values are zero at $t = 0$ and updated at every time step [16]. Temporal Difference (Equation (6)) updates Q-values by getting the difference between the reward received from taking an action that leads to the best known future action $r_t + \gamma \max_a(Q(a, s_{t+1}))$, and the current action played $Q(a_t, s_t)$. Temporal Difference (Equation (6)) is similar to an intrinsic reward. High is good and low is bad for the agent [16,22]. It can be formally defined as follows:

$$TD_t(a_t, s_t \ = \ r_t + \gamma \max_a(Q(a, s_{t+1})) - Q(a_t, s_t) \tag{6}$$

Q-Learning (Equations (7) and (8)) improves Temporal Difference (Equation (6)) by accumulating those high and low differences associated with each state-action pair. State-action pairs that have a high Temporal Difference are reinforced while those that are low are weakened. The Agent then learns through the Bellman Equation (Equation (8)), Q-values that will give it the maximum Temporal Difference [23]:

$$Q_t(a_t, s_t) \ = \ Q_{t-1}(a_t, s_t) + \alpha TD_t(a_t, s_t) \tag{7}$$

$$Q_t(s, a) = Q_{t-1}(s, a) + \alpha(R(s, a)\gamma Max_a Q(s', a') - Q_{t-1}(S, A)) \tag{8}$$

How the agent interprets Q-values is based on exploration-exploitation strategies defined earlier in this Section 2.1. Epsilon Greedy and Softmax are two popular methods with subtle differences [24]. Epsilon Greedy is a simple exploration method that exploits

actions with the highest estimated value most of the time but will randomly select other actions based on a small epsilon. Softmax exploration is more probabilistic where it assigns probabilities based on their Q-values. Epsilon Greedy is more binary, whereas in contrast Softmax (Equation (7)), considered in this paper, provides smoother exploration that accounts for the probability of all actions based on a Temperature *T* parameter. either exploring or exploiting [16].

$$W_s : a \in A \rightarrow \frac{\exp{(Q(s,a))}^T}{\sum_{a'} \exp{(Q(s,a'))}^T} ; T \geq 0 \tag{9}$$

The remaining Section 2.2 is devoted to Deep Q-Learning and its application, where classical tabular-based Q-Learning was replaced with a neural network Q-value approximator.

*2.2. Deep Q-Learning*

Deep Q-Learning (DQL), which uses the Bellman Equation (Equation (8)) [23] allows for control processes to be optimized in complex environments [17], such as playing computer games from pixels [11]. The DQL agent through trial and error learns the quality of taking an action in a given MDP state environment to find an optimal policy that maximises its total reward. The inputs to the neural network are states (vector of encoded values or an image as discussed in Section 2.1), and the outputs are Q-values. Using Epsilon Greedy or Softmax on the output layer converts Q-values into discrete actions. The Bellman Equation (Equation (8)) is used together with backpropagation to train the network. The ground truth $y$ in this case is the best known future q-value and the predicted q-value $\hat{y}$ is output by the neural network based on current state input. For vector data found in tabular or grid world RL can use a simple neural network to solve but for image-based worlds considered in this paper, a convolutional neural network is needed. The 1-dimensional flattened layer that represents features from the state image is passed to a neural network that outputs Q-Values, which Epsilon Greedy or Softmax use to select an action. When dealing with motion in image-based environments, a single image is not sufficient to ascertain the direction or speed of objects, a series of images is needed. When training an agent in simulation a series of n-steps in time $t$ are taken (e.g., this paper considers $n = 10$). Eligibility Trace can then be used for learning as it is a technique used to assign credit to past actions in a sequence of states and actions, to update the weights of a neural network model [25]. It helps address the problem of delayed rewards and enables more effective learning in environments with long-time delays between actions and their consequences. This is useful for learning to play complex games from pixels. During the learning process, the Eligibility Trace is updated at each n-step by decaying its value and adding the gradients of the weights with respect to the loss function. When an update is performed, the Eligibility Trace determines how much credit is assigned to each weight based on its previous contributions to the current outcome. By accumulating the Eligibility Traces over time, the algorithm can assign credit to actions that lead to future rewards, even if those rewards are delayed. Deep Reinforcement Learning (DRL) requires more data than regular deep learning [15]. Complex models can take many months to train with millions of iterations. Since it is impractical to collect data like this, DRL relies heavily on simulated environments. This causes a lot of issues when working with DRL. Some industries do not have access to simulated models of their processes. Simulations that do exist are often too simplistic causing the DRL model to overfit the simulation and fail when scaled to the real world. The solution here is calibrating the simulation to reflect reality as training and testing of machine learning models must follow the same distribution, but calibrating a simulation is costly and time-consuming, this is known as the "Sim2real" gap. Increasing the realism in simulation increases computational consumption making it difficult to quickly experiment and deploy DRL. Many simulations are not generic enough. A lot of commercial software is hard to integrate with DRL or may not be flexible enough to work with models [15] states that simulations should be fast, accurate, and scalable to many sessions. Simulated environments are vital in mimicking real-world problems safely, and generating valuable

training data [26]. This paper considers games for model training as they have defined rules and are complex enough to simulate real-world problems. DQL is used in computer games, autonomous driving, recommendation systems, robotics, energy grid optimization, fraud detection, pricing, and healthcare [1], where evaluation is comparing the performance of an Agent to a handcrafted, human expert or random policy. This paper evaluates the Agent against its past self at a different Experience Replay capacity size.

### 2.3. Experience Replay

Approximating Q-values with a neural network in large and complex states like autonomous driving, healthcare or computer games destabilizes learning, so [11,27] used Experience Replay [28] to sample data and store it from the environment for the approximator to later reuse. However, drawbacks included correlated samples, limited capacity causing an agent to forget information, outdated samples from non-stationary environments, and overfitting from samples memorized. Prioritized Experience Replay (PER) [29], Attention-Based Experience Replay [30], Combined Experience Replay (CER) [12], Explanation-Aware Experience Replay [31] and many more selection strategies [12–14,32] propose solutions to these drawbacks. Understanding Experience Replay is crucial for efficiency. Ref. [11] required 1 million frames of experience to outperform humans at 49 Atari games, Deepmind's state-of-the-art Agent57 [33], which beat human champions in Atari, and contained 80 billion frames of experience to achieve optimal performance. Consequently, many consider Experience Replay flawed and want to replace it due to the probability of a new transition being replayed monotonically decreasing as capacity increases [12]. Asynchronous Actor-Critic (A3C) by [27] is a popular alternative. It trains multiple agents in parallel to explore the environment and update a shared network, requiring more resources but converging faster. A general trend for deep learning has seen performance increase at the cost of an enormous resource expansion. For example, the deep image classification model AlexNet trained for 5 to 6 days on two GPUs in 2012 is surpassed by NASNet-A in 2018. NASNet-A cut the error rate in half but at the expense of 1000 times as much computing required [7]. Experience Replay, although slower, is more memory efficient, only requiring stored transitions and not multiple copies of the network. Ref. [12] highlighted that the size of Experience Replay $M$ is a neglected hyperparameter and, if large, hurts performance, stating that most defaults to 1M transitions used by [27] for the capacity size. Although [12] did not study how Experience Replay interacts with a Deep Convolutional Network, the authors proposed CER as a selection strategy to tackle the negative effect of a large buffer size and to be more efficient than PER. Ref. [13] investigated how the utility of different experiences is influenced by the control problem and proposed guidelines on how to use prior knowledge about the problem to choose an experience selection strategy. Ref. [13] recommended as a rule-of-thumb to keep Experience Replay high using 90% of total environment transition steps. Ref. [11] had 50 million frames of environmental transitions meaning instead of using 1 million [13] rule of thumb would suggest 45 million transitions. Ref.[14] showed in the Atari Learning Environment that increasing Experience Replay from 1 million to 10 million transitions while also decreasing the age of the oldest Policy did improve performance. However, reducing Experience Replay capacity risks data quality especially in rule-dense environments as [13,31] showed any increase in the size of Experience Replay further burdens resources [5,6].

### 2.4. Explainable Reinforcement Learning

Explainable Artificial Intelligence (XAI) aims to enhance the transparency and interpretability of AI systems, making their decision-making processes easily understandable. For Deep Reinforcement Learning models like Deep Q-Learning to become more prevalent, there is a need to improve the debugging and interpretation of their decision-making process. XAI is crucial for building trust, accountability, complying with regulations, and ethical considerations [34]. Various methods have been developed to achieve explainability in AI systems [34–36]. A common approach involves generating post-hoc explanations,

where models' decisions are explained after they have made predictions. Techniques like feature importance visualization, saliency maps, and SHAP (Shapley Additive exPlanations) fall into this category [34]. Researchers are exploring hybrid models that combine the power of complex models with interpretability layers. Methods such as TreeExplainer, a modified version of SHAP for tree-based ML models, and combining SHAP with the Lorenz Zonoids decomposition are used to determine relevant features by generating a receiver operating characteristic (ROC) curve. This paper considers Deep Explainer (Deep SHAP), which builds on the DeepLIFT (Deep Learning Important FeaTures) algorithm [9], a method to deconstruct the output prediction of a neural network based on a specific input by backpropagating the contributions of all neurons in the network to every feature of the input and SHapley Additive exPlanations (SHAP) [10]. SHAP recreates a model's outcome by quantifying the marginal contribution of features to that model for a single instance. For example, a salary predicting model could start with a base of €50 per hour and three feature inputs (age, gender, and experience) the model then outputs €40 per hour, as each feature is removed to determine its marginal contribution and it is found that age negatively impacts the model by −€10 per hour when compared to the base. The hybrid model Deep SHAP approximates SHAP values by creating connections with DeepLIFT and is trained on the distribution of base samples instead of a single reference value. In a Deep SHAP plot, the input image is presented in transparent grayscale while the output is presented as different colored images. Red pixels indicate an increase in the model's output while blue pixels decrease the output. The sum of the SHAP values equals the difference between the expected model output, which is averaged over the background images, and the current model output. In Deep Convolutional Q-Learning, this would be an image from a game and predicted Q-values for each action, n-steps into the future. A SHAP heat map could highlight why an action is obtaining a higher q-value, for example in SpaceInvaders, this could be due to the Agent expecting a mothership to appear causing the agent to favor shooting in an empty space to hit the ship and receive 300 points.

The minimum Experience Replay size allowed in DQL is unknown, but using Deep SHAP explainability while reducing it can provide insight into the behavior of the algorithm, training dynamics, or other relevant aspects, highlighting any unexpected observations or patterns to spark further interest or investigation [10,34–36]. Many custom explainers for DQL exist [37–40] to try to understand simulation events or the Agent within them, but not for Experience Replay samples that the model uses as training data to predict Q-values. Within Explainable Reinforcement Learning (XRL) [5,8,41], SHAP [10] is a popular choice for black-box explanation [39,40,42]. RL-SHAP diagrams exist to explain grid world environment features and their effect on action selection, but it has not been used for image-based simulations nor on samples stored in Experience Replay itself. Similarly, Experience Replay can be partitioned into clusters and given explainable labels based on rule density to select environment features [31], but this is an experience selection strategy and not a post-hoc explanation for debugging a DCQL model. This paper proposes the use of Deep SHAP Explanations to generate images of Experience Replay samples allowing patterns and trends to be visualized, to assist in debugging the DQL model as Experience Replay capacity is reduced. In detail, the following section describes a primary research experiment devoted to searching for a minimum Experience Replay in Deep Q-Learning and using Deep SHAP to visualize Experience Replay during the training process.

## 3. Design

A primary research experiment (Figure 1) is designed to test the effect of different Experience Replay capacities on the average cumulative reward. In this section data understanding, preparation, modeling, and evaluation related to the research experiment are described.

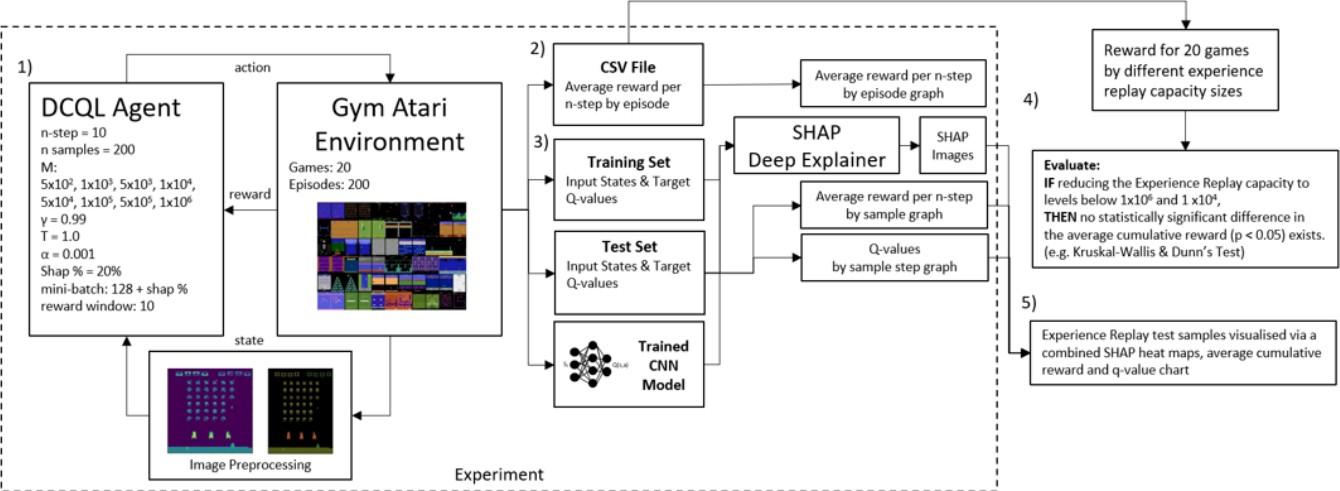

**Figure 1.** Experiment Process Design: (1) Run DCQL Agent inside the Atari environment to create a test dataset. (2) Save the dataset as a CSV file to later evaluate. (3) Extract unseen samples from Experience Replay and train the CNN model. Split samples into training and test sets. Train Deep SHAP Explainer. (4) Get a reward across 20 games and evaluate the Experience Replay size with the highest average cumulative reward. (5) Generate images from SHAP Deep Explainer and compare them to Q-values and rewards received during training.

### 3.1. Data Understanding

To obtain reward values by Experience Replay capacity an Agent comprising of a convolutional neural network receives state information and infers Q-values for each state. Actions are played to change the environment and update the state information. The environment used is called the *Atari Arcade Learning Environment*, which is integrated into *Open-ai gym* [26,43]. *Atari* games are simulated through *Stella* and mimic real-world problems to generate valuable training data in a secure manner. The 20 games considered as part of this paper are as follows; AirRaid, Asterix, Asteroids, Bowling, Breakout, DemonAttack, Freeway, Gravitar, Jamesbond, Montezuma Revenge, MsPacman, Pong, PrivateEye, Qbert, Seaquest, SpaceInvaders, Venture, WizardOfWor, Yar's Revenge and Zaxxon. Evaluation of performance is comparing the reward received by the Agent to its previous Policy, a handcrafted human expert Policy, or random Policy. In *Open-Ai Gym* each *Atari* game has a different set of states, actions and rewards. State information is provided as pixels. Actions are discrete values for example left or right move, stand still, or firing. Each game has a specific goal and reward structure. For example in SpaceInvaders a reward is given for every enemy shot down with a bonus reward for shooting the mother ship, in breakout a paddle and ball are used to remove bricks and in Montezuma Revenge, a reward is given for planning by navigating a castle collecting keys to unlock doors, stealing treasure and avoiding enemies.

### 3.2. Data Preparation

At each time step *t* the simulated environment will present the agent with state information and a reward. During the initial stages of the simulation the reward can be negative to trigger the agent to do something otherwise the agent may learn that staying stationary is the optimal policy to avoid a negative reward. As the agent proceeds to take action it can get new rewards both positive and negative, after each action the simulated environment will update with new state information for the agent. *Open-ai gym* provides a framework to receive data as an observation array during each step in time.

- Image States: Images provided to the Agent without prepossessing, cause training time to be long with no performance increase. Each image is reduced from its default size, similar to the original implementation by [11] (e.g., *Atari* SpaceInvaders: 210 px,

160 px, 3 color channels) to a greyscale image (i.e., 80 px, 80 px, 1 color). To ascertain motion and direction, images are collected and batched first then passed to the deep convolutional Q-learning model as discussed earlier in Section 2.2. Batching images is set to 10, meaning 10 steps are taken through time occurs then 10 images are grouped and stored in a history buffer to be later used. Ref. [11] used n-step = 4, Ref. [12] used n-step = 1, Ref. [14] experimented with n-step = 2 to n-step = 7.

- Discrete Actions: If the actions allowed are move-left, move-right, move-up or move-down then the Agent's discrete actions would be accepted as 4 by the environment. Each simulated environment has its own predefined number of actions. Some games in *Atari* can have up to 14 discrete actions.
- Data Sampling: 154 random samples are taken from the Agent's Experience Replay buffer as mini-batch samples. 128 are used as training data for the agent while the extra 20% is set aside as test data to later train a SHAP Deep Explainer.

Data prepossessing described in this section is fixed for every simulation. This data trains a convolutional neural network to approximate Q-values, which is described in the next section.

### 3.3. Modelling

The DCQL model described in this section replicates the original Deep Q-Learning architecture implemented by [11]. It can handle complex state information in pixel format as described previously in Section 2.2. The DCQL Agent's architecture is shown in Figure 2. It has 3 convolutional layers.

- The first layer takes a single $80\times80$ pixel image as an input feature and outputs 32 feature maps $5\times5$ pixels in size.
- The second layer takes 32 feature map images and outputs feature maps as $3\times3$ pixels in size.
- The final convolutional layer takes 32 feature maps and outputs 64 feature maps $2\times2$ in size.
- The Results of convolutions are passed into a fully connected neural network layer where the number of input neurons used is dynamically determined. This determination is first done by creating a blank fake black and white image (80 px $\times$ 80 px), max pooling the resulting layer through the 3 convolutional layers with a stride of 2 and flattening the layer into a one-dimensional layer. This results in the quantity of input neurons needed.

As stated previously in Section 2.2, A pooled layer is not included because the location of the objects inside an image needs to be preserved. A total of 40 hidden neurons are used, which are then passed to a second fully connected layer. Outputs are Q-values for the number of actions the Agent can play. These Q-values are explored or exploited by the Softmax Policy to select an action. The DCQL Agent is trained using Eligibility Trace, defined in Section 2.2. It is similar to Asynchronous n-step Q-learning used by [27], except Softmax is used instead of Epsilon Greedy, and Experience Replay is used instead of A3C as discussed in Section 2.2.

### 3.4. Evaluation

Finding optimal policies for *Atari* games may require significantly more steps. For example, when [11] trained a DCQL Agent in *Atari SpaceInvaders* to score higher than the best human players it took $10 \times 10^6$ to $40 \times 10^6$ steps. Given the hardware and time constraints, this paper set the number of episodes per game to 200, the same as [14], which allowed training to last an hour per simulation. An episode can contain many steps and last until a terminal state is triggered (i.e., time runs out or Agent loses the game). In the *Atari SpaceInvaders*, a terminal state is when the player is killed by alien invaders. From an informal search of each game, it was found to be about 2000 steps per episode. 200 sample states are then taken for every episode. A sample contains 10 time steps ($n = 10$).

This equates to a total of $4 \times 10^5$ steps where reward is received for taking an action. After a terminal state is reached the cumulative reward is recorded. A rolling average is calculated with a window size of 10 steps, similar to [14]. At the start of each game, the Experience Replay capacity is set to $1 \times 10^6$ and reduced to $5 \times 10^5$, $1 \times 10^5$, $5 \times 10^4$, $1 \times 10^4$, $5 \times 10^3$, $1 \times 10^2$, and $5 \times 10^2$ transitions respectfully. Once the reward is collected for each Experience Replay setpoint for each *Atari* game, it is saved to a CSV file. After all games have run each CSV file is merged together and an Analysis Of Variance (ANOVA) or its non-parametric equivalent Kruskal-Wallis test is subsequently executed, and a post-hoc test either Tukey or Dunn's test is run on the distribution of rewards. Deciding to use ANOVA over Kruskal-Wallis is based on a Shapiro-Wilk test confirming that reward data is normally distributed.

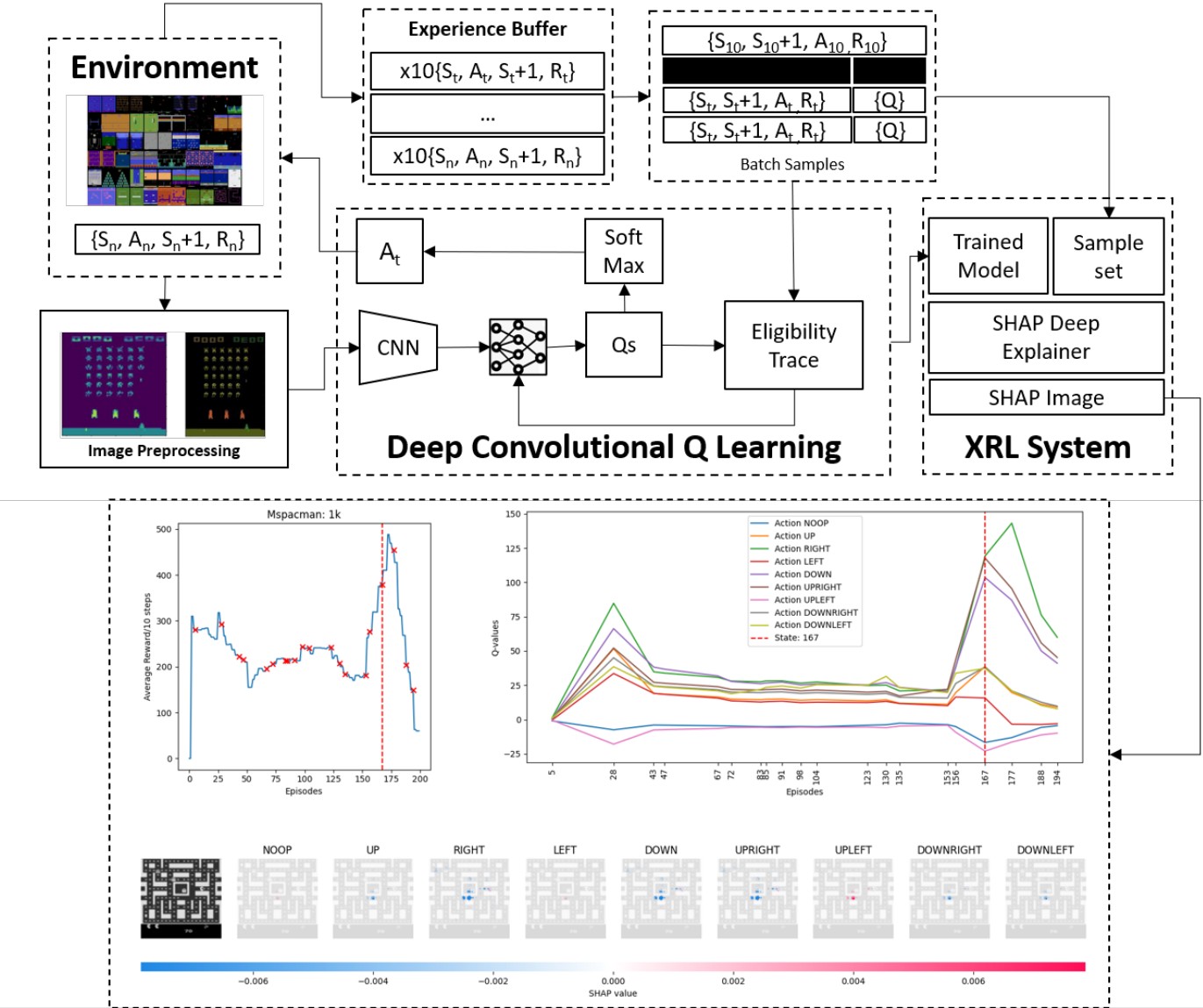

**Figure 2.** DCQL Architecture: (1) Agent takes 200 actions per episode. (2) Environment returns a reward + next state. 10 preprocessed image states are stored in an n-step history buffer. (3) Observations in experience replay are sampled 154 (128 + 26) times. (4) A neural network learns q-values outputting the next best action via Softmax. (5) Repeat for 200 episodes. 20 Experience samples extracted from buffer. (6) CNN model extracted. (7) Deep Explainer creates interpretable SHAP values.

This study investigates the effect of reducing the Experience Replay capacity below the initial setting of $1 \times 10^6$ used by [11] and the threshold of $1 \times 10^4$ (as explored in [12–14])

on the average cumulative reward in the context of Deep Q-Learning, specifically using 20 Atari games. This paper proposes that a reduction in Experience Replay capacity from either the initial setting or the threshold will not lead to a statistically significant difference in the average cumulative reward, with a significance level of $p < 0.05$, when compared to the baseline experience replay capacity of $1 \times 10^6$ and $1 \times 10^4$. Such a finding would suggest that the Experience Replay capacity hyperparameter $M$ can be effectively decreased without compromising performance. Formally we state this in detail:

> IF the Experience Replay capacity hyperparameter $M$ is reduced below $1 \times 10^6$ or below the $1 \times 10^4$, threshold investigated in [12–14], THEN we hypothesize that there exists a specific configuration of $M$ resulting in maximal reward scores while minimizing $M$, and this configuration demonstrates no statistically significant difference in average cumulative reward compared to the baseline $M$ values of $1 \times 10^6$ or $1 \times 10^4$, with a significance level of $p < 0.05$. This outcome would imply that the Experience Replay capacity hyperparameter $M$ can be decreased without causing a significant drop in performance.

The following specific statistical tests are used to test such research hypothesis:

- One-Way Analysis of Variance (ANOVA): This test is used to tell whether there are any statistically significant differences between the means of the independent (unrelated) groups of reward scores where experience replay capacity is set to $1 \times 10^6$ and reduced to $5 \times 10^5$, $1 \times 10^5$, $5 \times 10^4$, $1 \times 10^4$, $5 \times 10^3$, $1 \times 10^2$, and $5 \times 10^2$ transitions respectfully
- Shapiro-Wilk Test: This test is used to confirm if ANOVA and Tukey or Kruskal-Wallis test and Dunn's post hoc test can be used by checking if the reward data is in fact normally distributed, a prerequisite for ANOVA [44].
- Kruskal-Wallis Test: This is based on the ranks of the data rather than the actual values. It ranks the combined data from all groups and calculates the test statistic, which similar to ANOVA, can measure the differences between the ranked group medians. The test statistic follows a chi-squared distribution with $(k - 1)$ degrees of freedom, where k is the number of groups being compared. The null hypothesis of the Kruskal-Wallis test is that there are no differences in medians among Experience Replay size groups. The alternative hypothesis suggests that at least one group differs from the others [45].
- Tukey Test: This test is used after an ANOVA test. Since, ANOVA can identify if there are significant differences among group means, a Tukey test can identify which specific pairs of group means are significantly different from each other.
- Dunn's Test: This test is similar to Tukey but used after a Kruskal-Wallis test and determines which Experience Replay groups have significantly different sizes.

## 4. Results and Discussion

### 4.1. Finding Minimum Experience Replay Allowed

Figure 3 shows the first group ($5 \times 10^5$, $1 \times 10^5$, $5 \times 10^4$), compared to the baseline $1 \times 10^6$, and second group ($5 \times 10^3$, $1 \times 10^2$, $5 \times 10^2$) compared to $1 \times 10^4$. Outliers are valid points from the 20 Atari games, signifying the rewards from games with higher rewards than others. Focusing on the median and interquartile ranges, there seems to be a difference in medians between $1 \times 10^6$ and other groups. Interestingly however $5 \times 10^3$ and $1 \times 10^4$ have the same mean, warranting further statistical testing. As discussed in Section 3, full details of individual games can be found in Appendix A. The following describes the results of the Shapiro-Wilk Test, Kruskal-Wallis, and Dunn's Test. Shapiro-Wilk test ($p < 0.001$) shows ANOVA assumptions void, however, a Kruskal-Wallis Test ($p < 0.001$) finds that there is a substantial difference among groups. Table 1, shows the results of a Dunn's Post Hoc Test, where rows represent source groups, and column represents target groups. Cells are p-values for the pairwise comparisons between the source and target groups. Diagonal values equal one due to the source and target groups being the same.

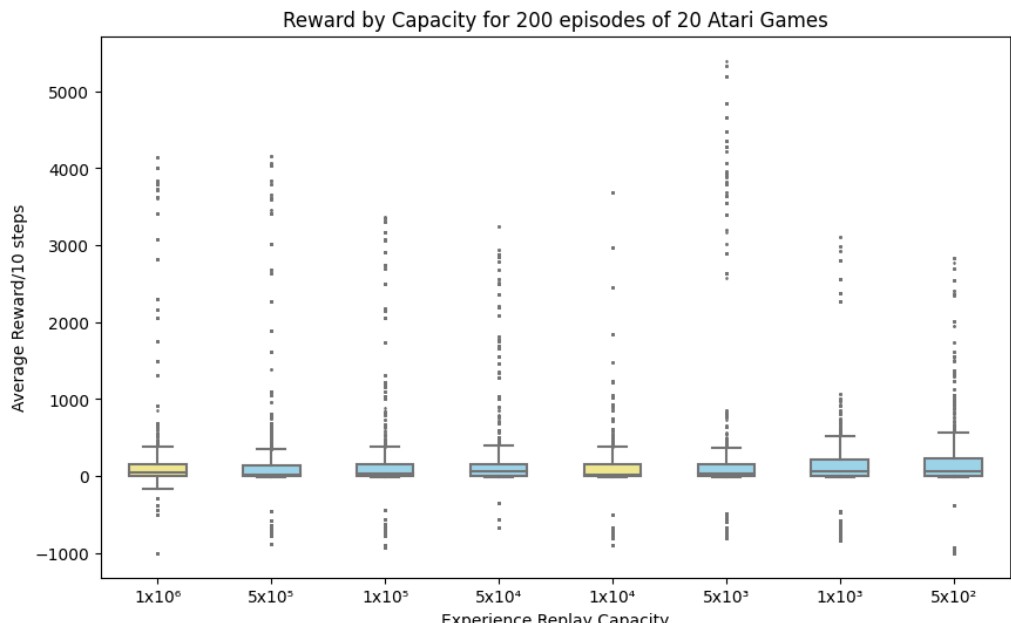

**Figure 3.** Reward by Capacity: $1 \times 10^6$ and $1 \times 10^4$ baseline compared to first group ($5 \times 10^5$, $1 \times 10^5$, $5 \times 10^4$) and second group ($5 \times 10^3$, $1 \times 10^2$, and $5 \times 10^2$).

**Table 1.** Dunn's Post Hoc Test.

|  | $1 \times 10^6$ | $5 \times 10^5$ | $1 \times 10^5$ | $5 \times 10^4$ | $1 \times 10^4$ | $5 \times 10^3$ | $1 \times 10^3$ | $5 \times 10^2$ |
|---|---|---|---|---|---|---|---|---|
| $1 \times 10^6$ | **1.000** | **0.001** | **0.111** | **0.017** | **0.003** | **0.003** | <0.001 | <0.001 |
| $5 \times 10^5$ | **0.001** | **1.000** | **0.090** | **0.715** | **0.722** | <0.001 | <0.001 | <0.001 |
| $1 \times 10^5$ | **0.111** | **0.090** | **1.000** | <0.001 | **0.181** | **0.184** | <0.001 | <0.001 |
| $5 \times 10^4$ | **0.017** | **0.715** | <0.001 | **1.000** | <0.001 | <0.001 | **0.009** | **0.001** |
| $1 \times 10^4$ | **0.003** | **0.722** | **0.181** | <0.001 | **1.000** | **0.993** | <0.001 | <0.001 |
| $5 \times 10^3$ | **0.003** | <0.001 | **0.184** | <0.001 | **0.993** | **1.000** | <0.001 | <0.001 |
| $1 \times 10^3$ | <0.001 | <0.001 | <0.001 | **0.009** | <0.001 | <0.001 | **1.000** | **0.480** |
| $5 \times 10^2$ | <0.001 | <0.001 | <0.001 | **0.001** | <0.001 | <0.001 | **0.480** | **1.000** |

Based on the Dunn's Post Hoc Test in Table 1 the value at the intersection of $1 \times 10^4$ and $5 \times 10^3$ is approximately $p = 0.993$. The value at intersections of matching rows and columns equals 1.000, indicating no significant difference (as expected). This means that the *p*-value for the comparison between the groups with Experience Replay capacity of 10,000 and 5000 is close to 1, indicating no significant difference between these groups of average cumulative reward for 20 Atari games.

A DCQL Agent, earlier described in Section 3, is placed in 20 Atari games. A Kruskal-Wallis test and Dunn's post hoc test are used due to reward data failing the Shapiro-Wilk test of normality for ANOVA. The null hypothesis for the Kruskal-Wallis test is rejected. There is a difference ($p < 0.05$ in reward scores when Experience Replay is reduced. However, a post hoc Dunn's Test revealed the smallest Experience Replay allowed in 20 Atari Games is $5 \times 10^3$ (Table 1). As Figure 4 revealed the DCQL model was too simplistic to score high and some games proved too challenging to get any suitable results (Montezuma's Revenge and Venture). Experience Selection based on random uniform sampling is not an optimal selection strategy [12–14,29–32] and as Experience Replay is reduced there is a risk that rule dense features will be lost however capacity can be reduced from $1 \times 10^4$ to $5 \times 10^3$ without significantly affecting reward. This in turn will improve efficiency by reducing the impact on resources.

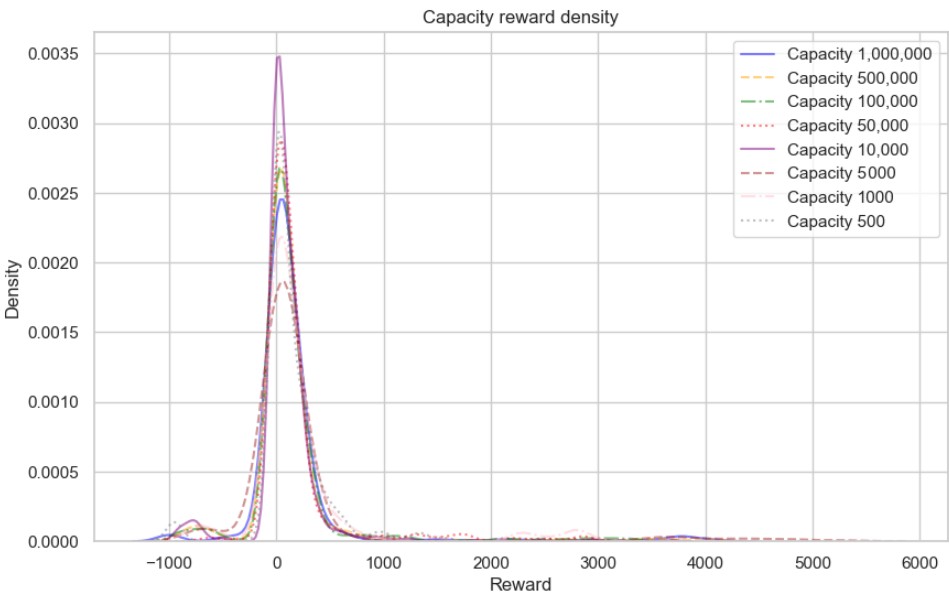

**Figure 4.** Density Plot: Probability density of reward for each capacity size is close to zero indicating that the model is not scoring high in each game. Experience selection based on uniform random sampling is not an optimal selection technique and n-step = 10 or softmax T = 1.0 may be too high but interestingly a size of $5 \times 10^3$ returns higher rewards than $1 \times 10^4$.

The remainder of this section discusses the results of visualizing Experience Replay with Deep Shap Explainer. Images containing SHAP values were investigated to discover if the Agent's performance on a minimum setting is in fact working, as in the Agent has learned the underlying concepts of the game and what outliers may be skewing the result. As stated in Section 1, images are not being presented as support for the hypothesis, only as a visual illustration of the outcomes and trends observed during the training process. The 20 games played by the DCQL agent were as follows: AirRaid, Asterix, Asteroids, Bowling, Breakout, DemonAttack, Freeway, Gravitar, Jamesbond, Montezuma Revenge, MsPacman, Pong, PrivateEye, Qbert, Seaquest, SpaceInvaders, Venture, WizardOfWor, Yar's Revenge and Zaxxon. Focus is given to MsPacman.

### 4.2. Visualising Experience from MsPacman

In this instance in Figure 5, a capacity size of 1 million or 100,000 results in scores higher than others. The only exception is a capacity size of 1000. Here the DCQN Agent seems to be exploiting rules with newer experience than storing and replaying older experience. Using SHAP we can examine sample experiences from capacity size 1000, at points of interest along the training to see why the DCQN is taking the actions it did. Two particular points of interest are episode 167 (Figure 6) and episode 177 (Figure 7).

Based on the reward graph in Figure 6, previous actions have been successful in increasing reward. Red crosses on the reward chart indicate where random samples of Experience replay were taken for Deep Shap Explainer. The vertical dashed red line indicates that episode 167 is being viewed. The graph on the right indicates Q-values being adjusted over time as the DCQL agent tries to learn from the experience collected from its environment. The bottom in Figure 6 represents calculated SHAP values overlayed on images based on a reference state image. Starting with this background box it indicates the DCQN agent started in the middle and moved up to the right. Each of the 9 actions in this game is presented. Blue indicates that these pixels are negatively influencing the Q-value prediction and red indicates a positive influence. The intensity of these colors signifies the magnitude of this influence, the result of which can be seen from the next lines on the Q-value chart. Remembering back to Section 2.2, Q-values are explored and exploited by the Softmax policy, a high Q-value may not necessarily be played based on the level of *T*

set prior to running the game but an increase in a Q-value ensures a higher chance of that action being played.

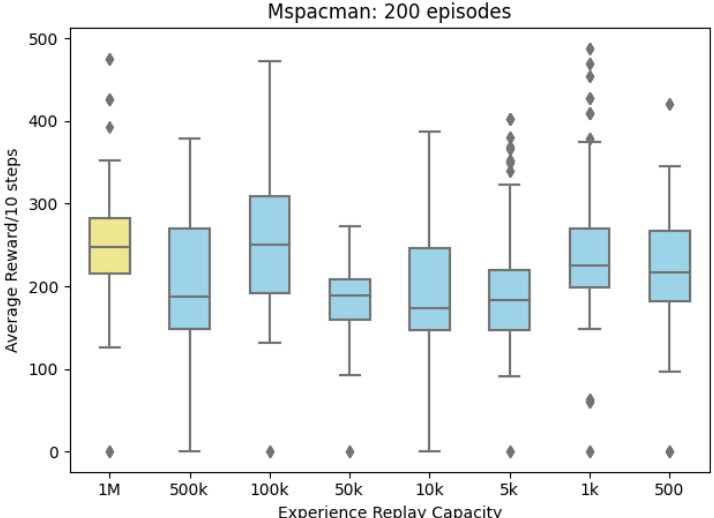

**Figure 5.** MsPacman Boxplot: Rewards received from MsPacman after reducing Experience Replay for 200 episodes each.

Using the UPLEFT action in 7th image from the left of Deep SHAP visualization as an example, in Figure 6, we can see that it was being penalized by the model but now is given priority. The Agent is replaying its experience of moving from bottom left to middle right, it is considered continuing left but both enemies are moving to the left opening up room for the the agent to move unharmed. However a small tint of blue can be seen where the Agent is, this is signifying that it wants to reinforce its reward of collecting pellets to the right so these pixel inputs are slightly reducing the Q-value output prediction.

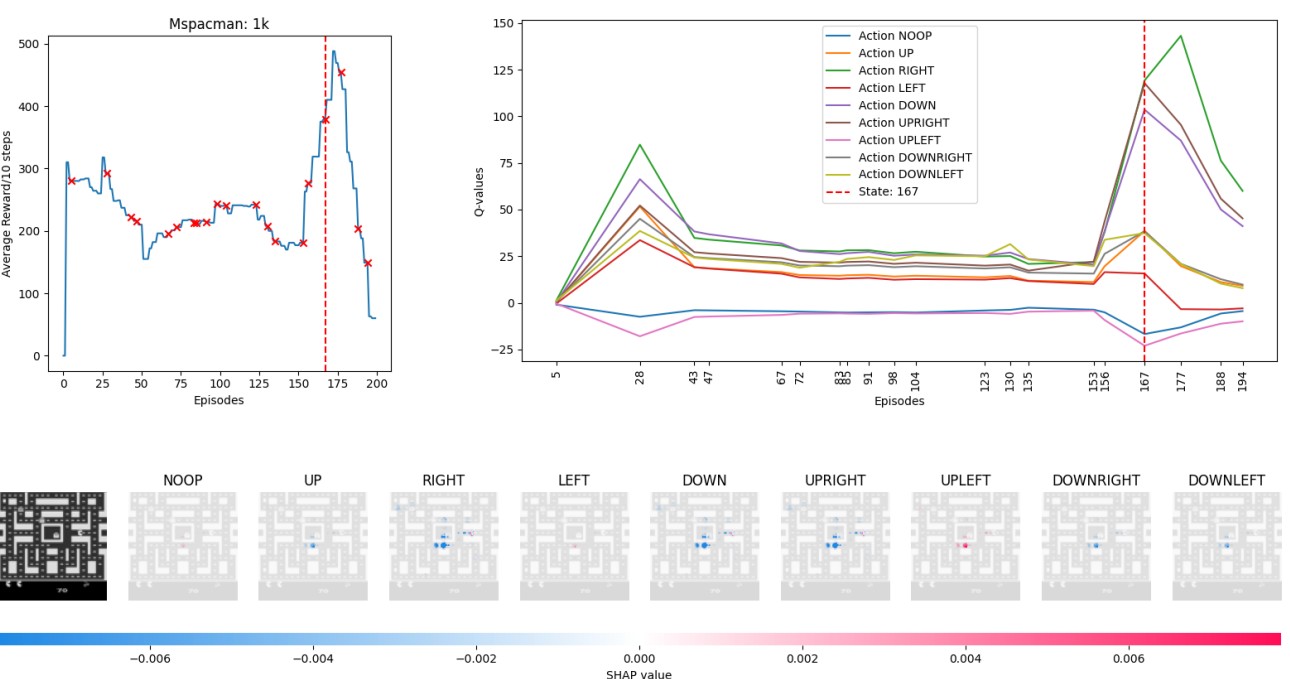

**Figure 6.** MsPacman Episode 167: Moments before a large reward received. Red crosses on the reward chart indicate where random experience samples were taken for Deep SHAP training. The vertical dashed red line indicates the episode being displayed by Deep SHAP.

Inspecting Figure 7, we now see that the agent received a large increase in reward from previous episodes. We see from the Q-value chart that the Agent is biasing its action selection to move toward the right of the screen. All actions were penalized except RIGHT, NOOP (Do nothing), and UPLEFT. Past experiences of dying are being replayed (bright red middle semi-circle on UPLEFT image, small red tint in NOOP), and pixels related to this are causing the agent to award value to standing still and moving to the upper left of the screen. Since RIGHT has a higher Q-value than other actions there is a higher probability that it will be played on the next turn but enemy movement and a previous death are negatively influencing a reduction in this action's Q-value. The Agent has grasped the underlying concepts of MsPacman but is biasing towards moving to the upper left of the screen. We conjecture that the observed dark blue SHAP values are highlighting past enemy positions that the agent thinks they will not be in future episodes, by the sudden decrease in reward received after episode 177 this seems to be a fatal mistake and could have possibly been avoided with a larger Experience Replay buffer size to hold older experiences or from a correct Experience Selection technique. Visualization using Deep SHAP Explainer provides a way to interpret the agent's decision-making process and validate whether its actions are aligned with the game's mechanics and objectives. This visualization adds an interpretative layer to the statistical testing in Section 4 and helps highlight that Experience Replay capacity can be reduced but should be done so with caution as the risk of biasing the agent increases, especially if the wrong Experience Selection strategy is chosen and rule density is not considered. By using SHAP values, we can visualize the effect of different Experience Replay capacities. In future work, it may be possible to show that if the agent's decisions are consistent and aligned with the game's mechanics and objectives while capacity is reduced, visualization can help support a reduction.

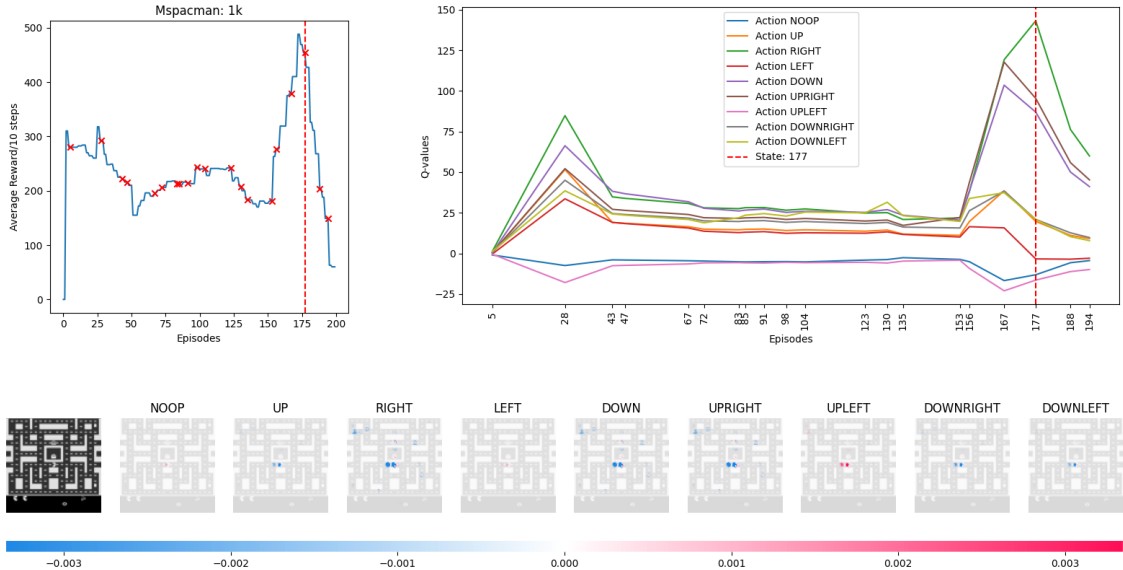

**Figure 7.** MsPacman Episode 177: shows rewards received from MsPacman after reducing Experience Replay for 200 episodes each. Red crosses on the reward chart indicate where random experience samples were taken for Deep SHAP training. The vertical dashed red line indicates the episode being displayed by Deep SHAP. In this instance, a capacity size of 1 million or 100,000 results in scores higher than others. The only exception is a capacity size of 1000. Here, the DCQN Agent seems to be exploiting rules with newer experience than storing and replaying older experience.

## 5. Conclusions

Reinforcement Learning (RL) holds the potential for optimizing complex control and decision-making processes [1]. However, its limited explainability, especially in Deep Reinforcement Learning (DRL), has hindered its adoption in regulated sectors like manufac-

turing [2], finance [3], and health [4]. These challenges, including debugging, interpretation, and inefficiency, become more pronounced as Deep Learning advances, such as in improving Experience Replay [12–14,29–32]. While many have focused on improving Experience Replay's selection strategy, reducing its capacity remains largely unexplored. In this study, we trained a Deep Convolutional Q-Learning agent in 20 Atari games, reducing Experience Replay capacity from $1 \times 10^4$ to $5 \times 10^3$, showing it can be done without significant impact on rewards. We also show that visualizing Experience Replay using Deep SHAP Explainer provides a way to interpret the agent's decision-making process and validate whether its actions are aligned with the game's mechanics and objectives. Using the Atari game MsPacman as an example, in Section 4.2 we show visualization helps highlight that Experience Replay capacity can be reduced but should be done so with caution as the risk of biasing the agent to overfitting, especially if the wrong Experience Selection strategy is chosen and rule density is not considered. In future studies, we plan to build upon our findings and explore further avenues for enhancing the efficiency and interpretability of Deep Reinforcement Learning (DRL) models. Specifically, we aim to extend our experiment by implementing it in Dopamine and testing different algorithms such as Rainbow, with variations in n-steps and experience retention ratios, as demonstrated in prior research [12–14]. Additionally, we intend to investigate alternative selection strategies, including the intriguing Explanation-Aware Experience Replay proposed by [31]. While our current study has shown that Experience Replay capacity can be reduced without significant impacts on rewards, it is essential to approach capacity reduction cautiously, as any decrease in capacity risks overfitting. By addressing these aspects, we aim to contribute to the ongoing effort to improve the resource efficiency and interpretability of DRL models.

**Author Contributions:** Conceptualisation, R.S.S. and L.L.; Supervision, L.L.; Writing—original draft, R.S.S.; review and editing, R.S.S. and L.L. All authors have read and agreed to the published version of the manuscript.

**Funding:** This research received no external funding.

**Institutional Review Board Statement:** Not applicable.

**Informed Consent Statement:** Not applicable.

**Data Availability Statement:** Datasets for experiments were generated from simulating environments in OpenAi Gym. This data can be recreated by implementing the algorithms in the same 23 environments. The corresponding authors can be contacted for any further clarification needed.

**Conflicts of Interest:** The authors declare no conflict of interest and have no relevant financial interest in the manuscript.

## Appendix A. Boxplots of Reward

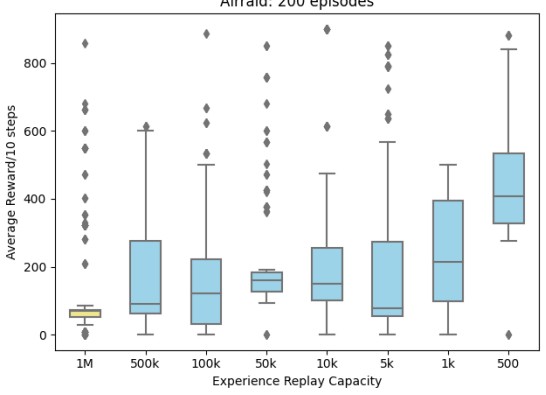
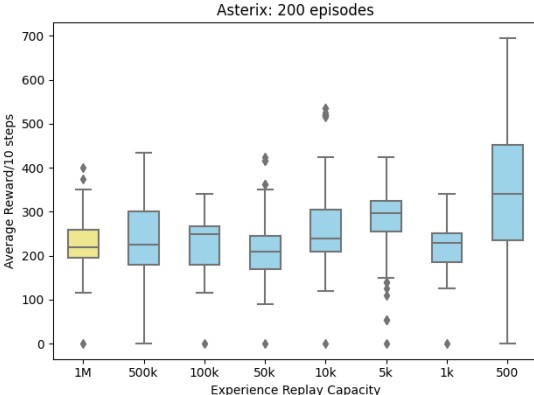

**Figure A1.** *Cont.*

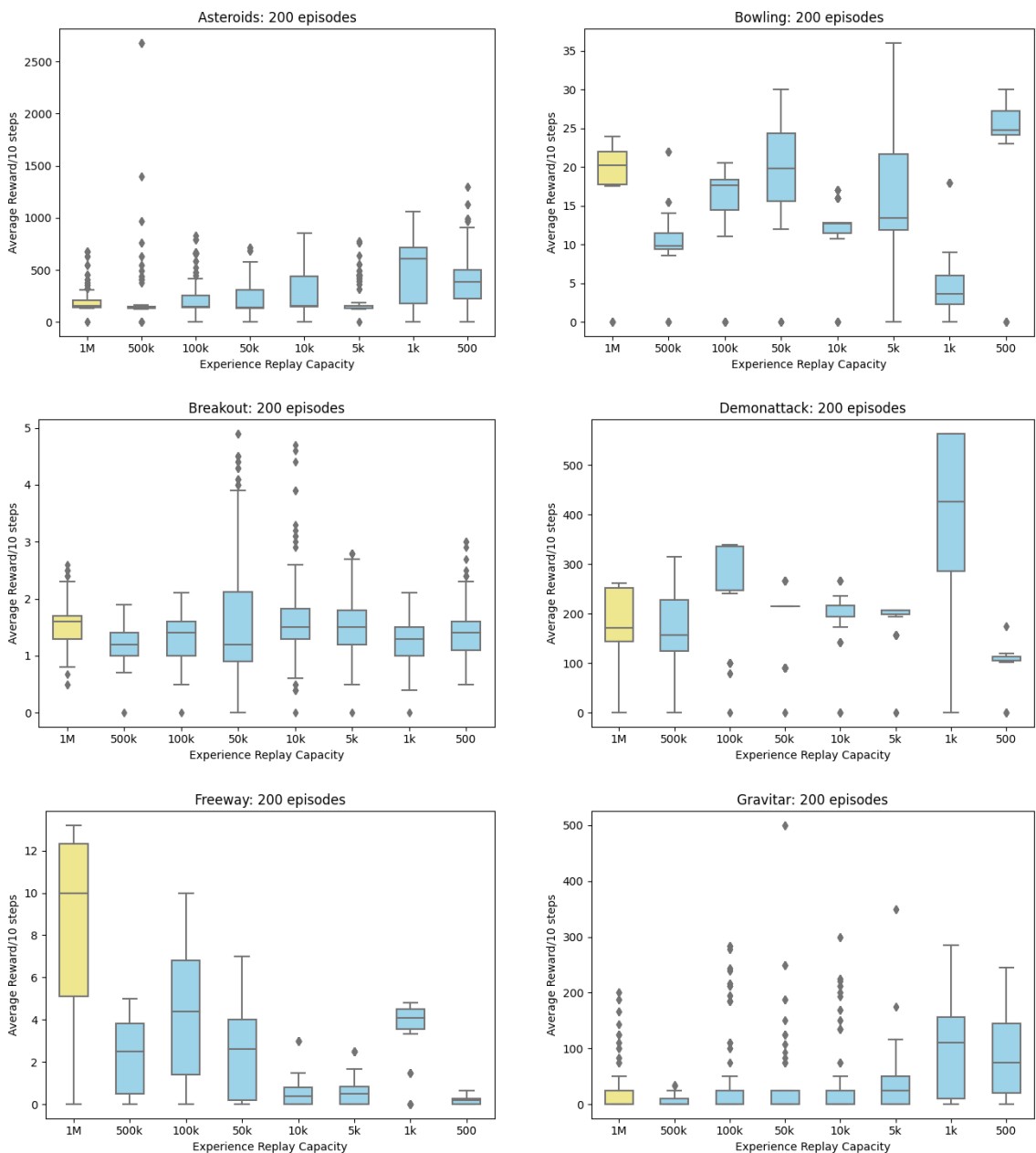

**Figure A1.** Boxplot Rewards for 20 Atari Games.

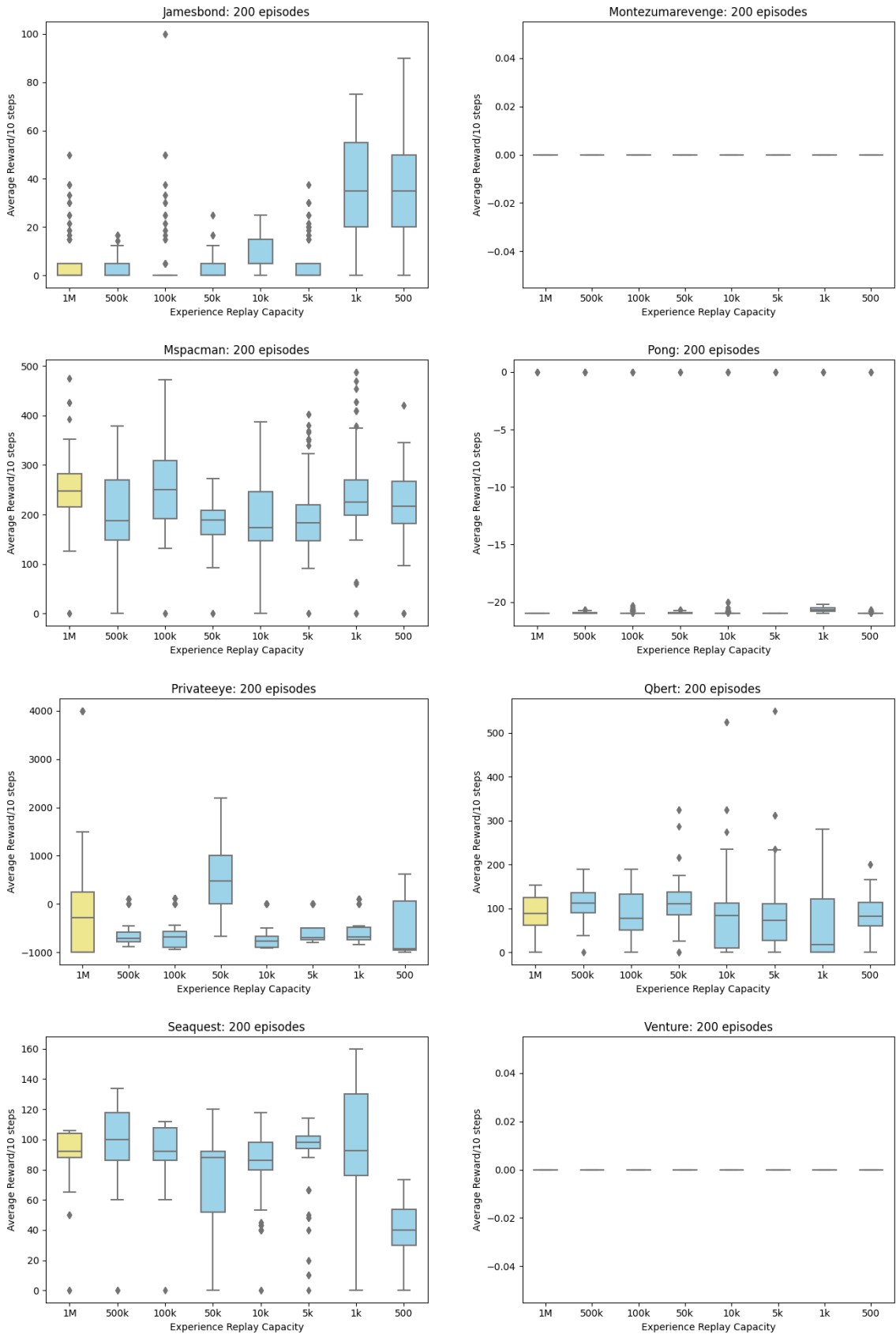

**Figure A2.** Boxplot Rewards for 20 Atari Games.

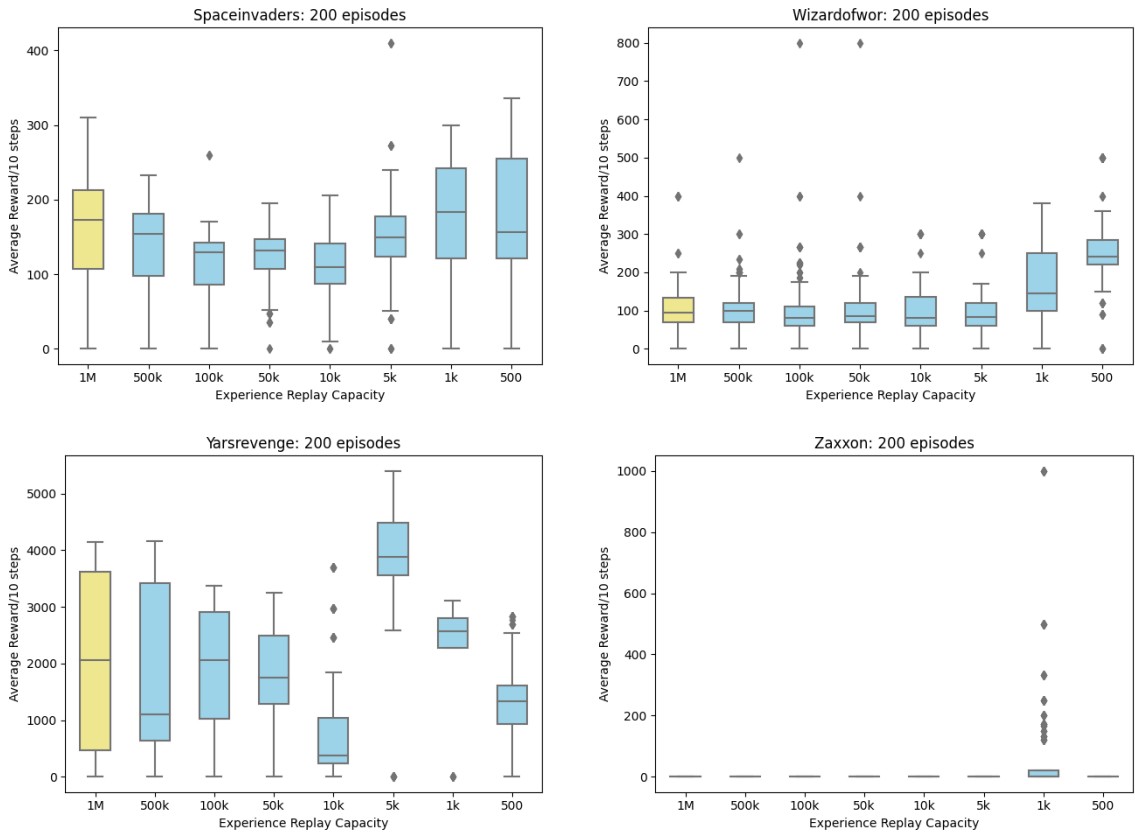

**Figure A3.** Boxplot Rewards for 20 Atari Games (continued).

## Appendix B. Line Charts of Reward

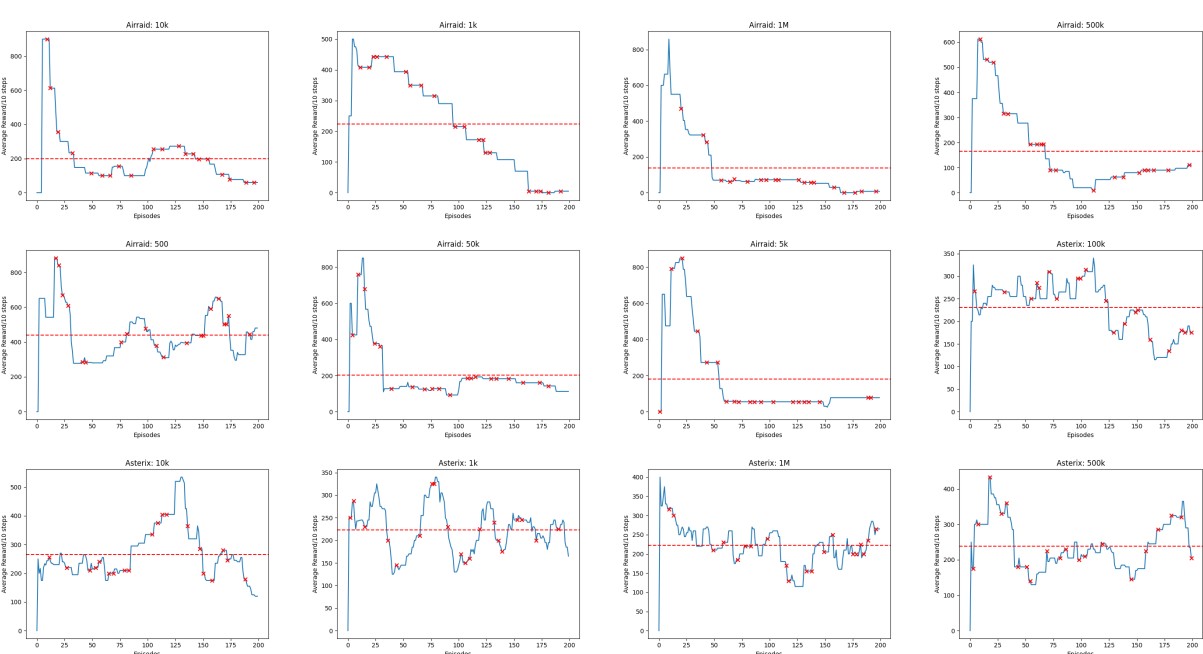

**Figure A4.** *Cont.*

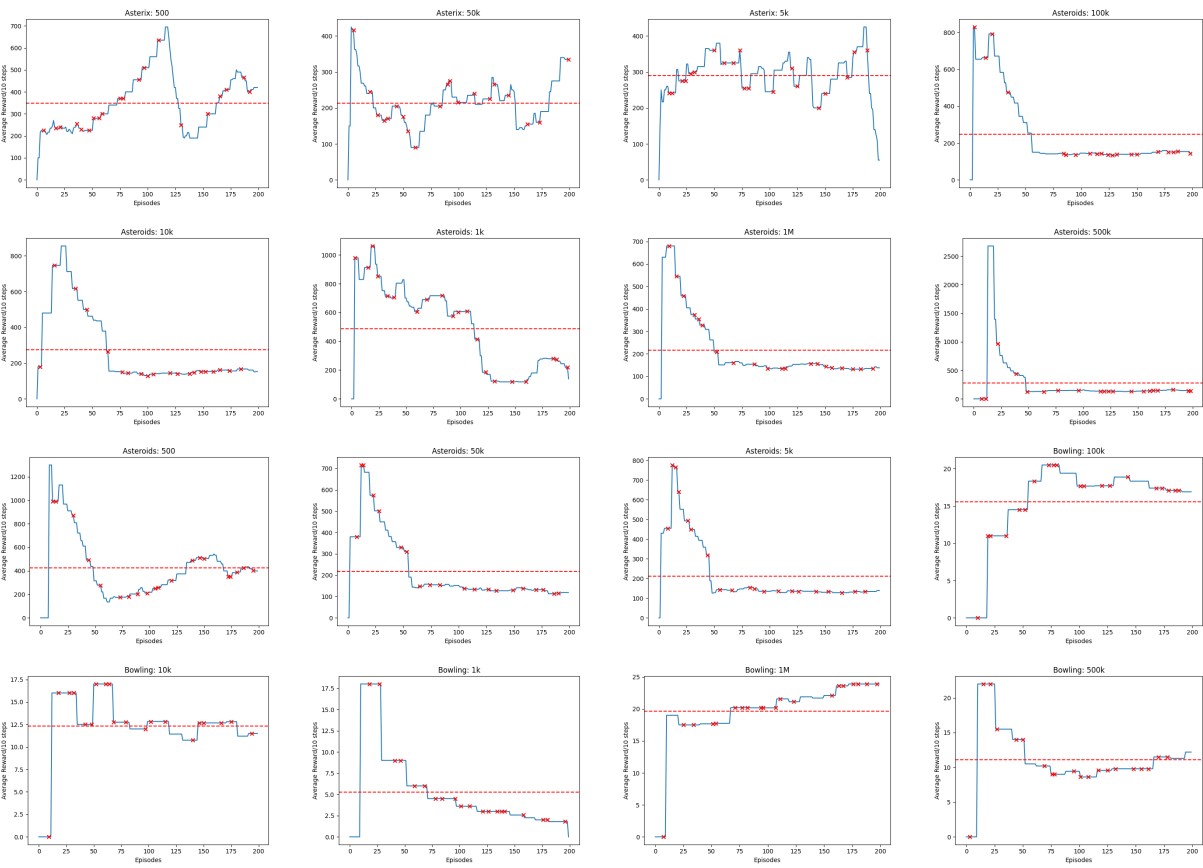

**Figure A4.** Linechart Rewards for 20 Atari Games. The red dashed horizontal line indicates the average cumulative reward received by the Agent.

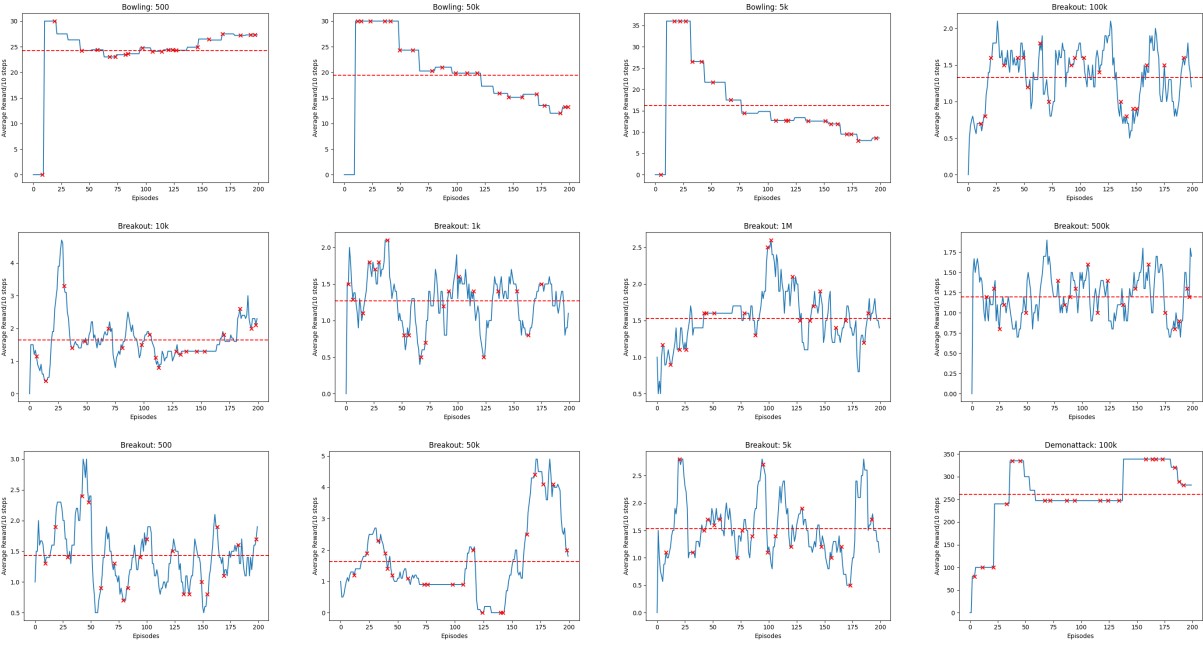

**Figure A5.** *Cont.*

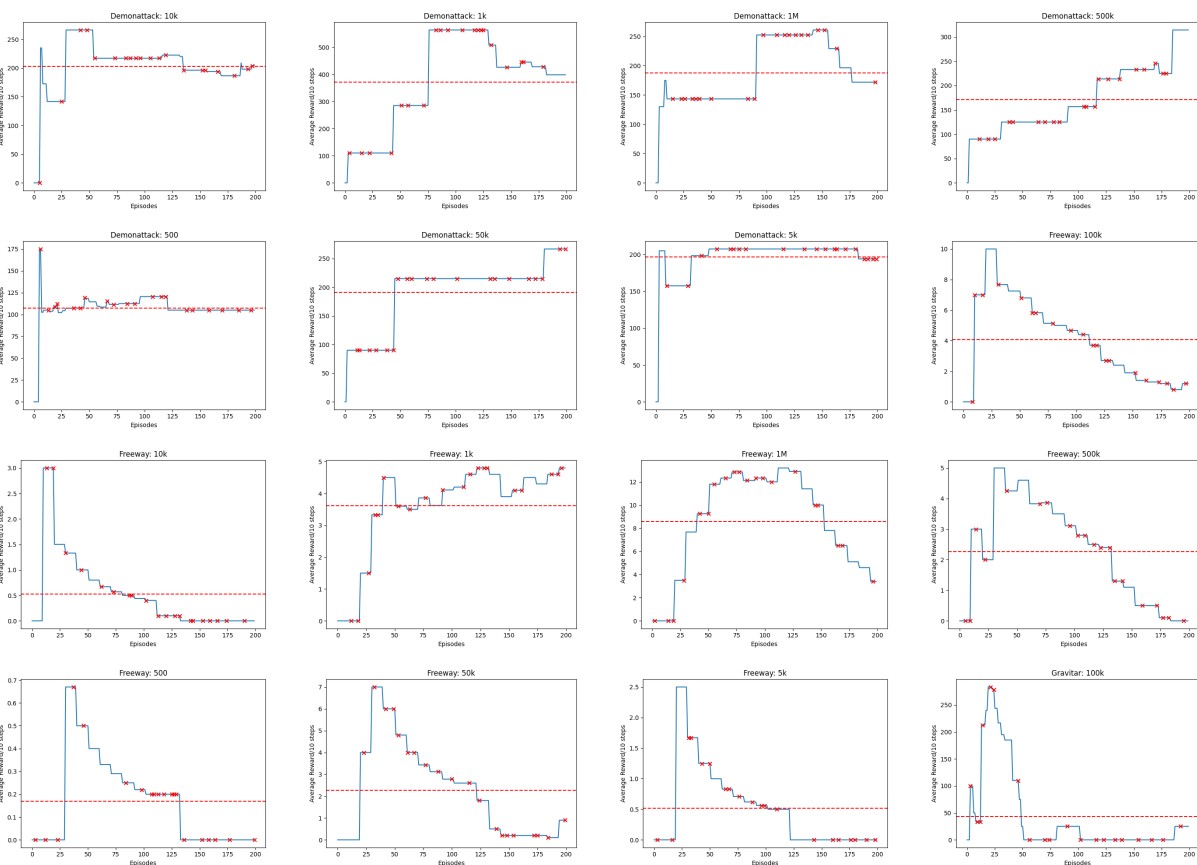

**Figure A5.** Linechart Rewards for 20 Atari Games (continued). The red dashed horizontal line indicates the average cumulative reward received by the Agent.

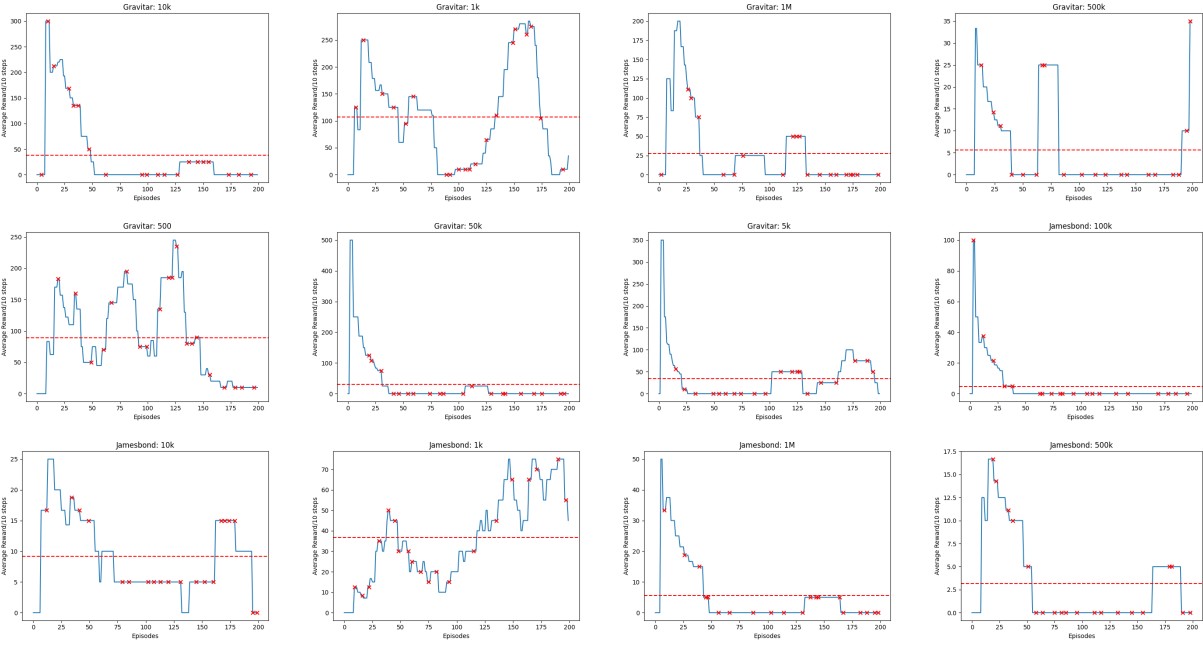

**Figure A6.** *Cont*.

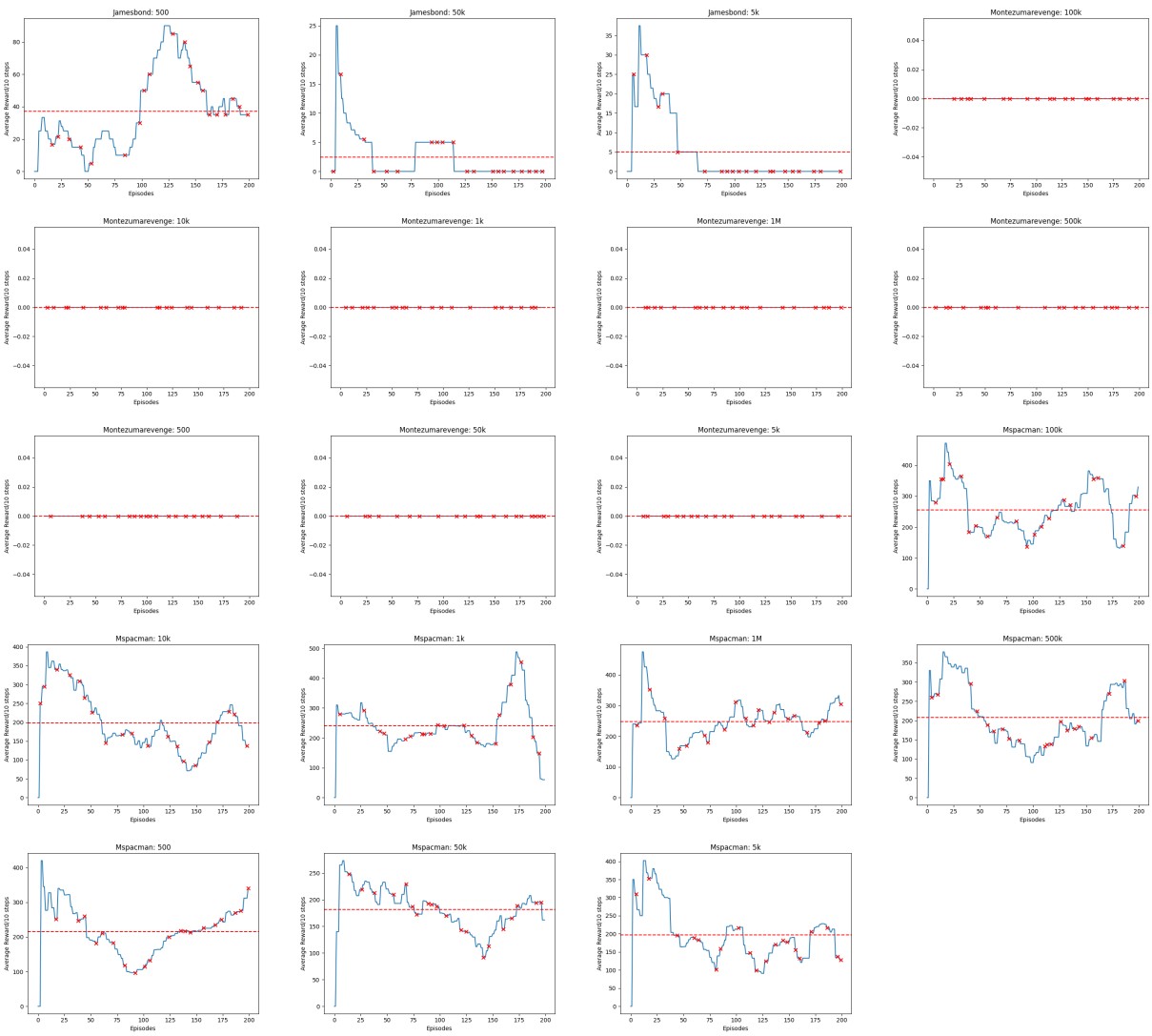

**Figure A6.** Linechart Rewards for 20 Atari Games (continued). The red dashed horizontal line indicates the average cumulative reward received by the Agent.

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
