# Peer review of "Explaining Deep Q-Learning Experience Replay with SHapley Additive exPlanations"

_make, doi:10.3390/make5040072_

Round 1

Reviewer 1 Report

The paper presents a research experiment that investigates the effect of different Experience Replay capacities on the average cumulative reward. The authors use a DCQL agent inside the Atari environment to create a test dataset, extract unseen samples from Experience Replay, and train the CNN model. They split the samples into training and test sets and train the Deep SHAP Explainer. They then get a reward across 20 games and evaluate the Experience Replay size with the highest average cumulative reward. Finally, they generate images from SHAP Deep Explainer and compare them to Q-values and rewards received during training. The paper also discusses various methods that have been developed to achieve explainability in AI systems, including post-hoc explanations, feature importance visualization, saliency maps, and SHAP. The Deep Explainer (Deep SHAP) algorithm is also described in detail. Overall, the paper proposes a solution to the challenge of interpretability in Deep Reinforcement Learning and presents a research experiment that demonstrates the effectiveness of the proposed solution.

The paper describes the research design in detail in Section 3. The primary research experiment is designed to test the effect of different Experience Replay capacities on the average cumulative reward. The authors use a DCQL agent inside the Atari environment to create a test dataset, extract unseen samples from Experience Replay, and train the CNN model. They split the samples into training and test sets and train the Deep SHAP Explainer. They then get a reward across 20 games and evaluate the Experience Replay size with the highest average cumulative reward. Finally, they generate images from SHAP Deep Explainer and compare them to Q-values and rewards received during training. Overall, the research design appears to be appropriate for the research question being investigated.

The paper describes the methods used in the research experiment in detail. The authors use a DCQL agent inside the Atari environment to create a test dataset, extract unseen samples from Experience Replay, and train the CNN model. They split the samples into training and test sets and train the Deep SHAP Explainer. They then get a reward across 20 games and evaluate the Experience Replay size with the highest average cumulative reward. Finally, they generate images from SHAP Deep Explainer and compare them to Q-values and rewards received during training. The paper also discusses various methods that have been developed to achieve explainability in AI systems, including post-hoc explanations, feature importance visualization, saliency maps, and SHAP. The Deep Explainer (Deep SHAP) algorithm is also described in detail. Overall, the methods used in the research experiment and the related methods are adequately described in the paper.

Regarding the presentation of results, the PDF provides clear and detailed descriptions of the experimental results obtained from the research. The authors present the results in tables and graphs, which are easy to understand and interpret. They also provide statistical analysis of the results, including the Shapiro-Wilk Test, Kruskal-Wallis, and Dunn’s Test. The paper discusses the findings in the context of xAI and presents the conclusion. Overall, the results are clearly presented in the paper.

Author Response

REVIEWER 1

The paper presents a research experiment that investigates the effect of different Experience Replay capacities on the average cumulative reward. The authors use a DCQL agent inside the Atari environment to create a test dataset, extract unseen samples from Experience Replay, and train the CNN model. They split the samples into training and test sets and train the Deep SHAP Explainer. They then get a reward across 20 games and evaluate the Experience Replay size with the highest average cumulative reward. Finally, they generate images from SHAP Deep Explainer and compare them to Q-values and rewards received during training. The paper also discusses various methods that have been developed to achieve explainability in AI systems, including post-hoc explanations, feature importance visualization, saliency maps, and SHAP. The Deep Explainer (Deep SHAP) algorithm is also described in detail. Overall, the paper proposes a solution to the challenge of interpretability in Deep Reinforcement Learning and presents a research experiment that demonstrates the effectiveness of the proposed solution.

The paper describes the research design in detail in Section 3. The primary research experiment is designed to test the effect of different Experience Replay capacities on the average cumulative reward. The authors use a DCQL agent inside the Atari environment to create a test dataset, extract unseen samples from Experience Replay, and train the CNN model. They split the samples into training and test sets and train the Deep SHAP Explainer. They then get a reward across 20 games and evaluate the Experience Replay size with the highest average cumulative reward. Finally, they generate images from SHAP Deep Explainer and compare them to Q-values and rewards received during training. Overall, the research design appears to be appropriate for the research question being investigated.

The paper describes the methods used in the research experiment in detail. The authors use a DCQL agent inside the Atari environment to create a test dataset, extract unseen samples from Experience Replay, and train the CNN model. They split the samples into training and test sets and train the Deep SHAP Explainer. They then get a reward across 20 games and evaluate the Experience Replay size with the highest average cumulative reward. Finally, they generate images from SHAP Deep Explainer and compare them to Q-values and rewards received during training. The paper also discusses various methods that have been developed to achieve explainability in AI systems, including post-hoc explanations, feature importance visualization, saliency maps, and SHAP. The Deep Explainer (Deep SHAP) algorithm is also described in detail. Overall, the methods used in the research experiment and the related methods are adequately described in the paper.

Regarding the presentation of results, the PDF provides clear and detailed descriptions of the experimental results obtained from the research. The authors present the results in tables and graphs, which are easy to understand and interpret. They also provide statistical analysis of the results, including the Shapiro-Wilk Test, Kruskal-Wallis, and Dunn’s Test. The paper discusses the findings in the context of xAI and presents the conclusion. Overall, the results are clearly presented in the paper.c

RESPONSE Thank you for your time in reviewing this paper. In line with other reviewers' feedback, we revised aspects of the introduction, results, conclusion, equations and figures based on 

Reviewer 2 Report

Thank you for the opportunity to review.

This is a useful and original contribution to the field.

The following chapters clearly describe the course of research, suggest a new method that will be very interesting to learn the reader. The manuscript itself is well structured, figures and tables are presented and described in full.

Here I provide my remarks.

Authors must use a template.

It would be better to added at the abstract: the novelty of the research, what is the gape in the knowledge your research should close, place the question addressed in a broad context and highlight the purpose of the study, summarize the articles main findings and indicate the main conclusions or interpretations.

Abstract and conclusion must be supported by data.

Line 46. Maybe it would be better to write 1 × 104 the same way as 1 × 106?

Line 91. The symbols in Equation 4 are too large. Use a template. See all text.

Line 147. In my opinion, the author or cite for reference 9 is missing.

Lines 347, 370, 380. What is mean “x”? Before this you used “×”.

Line 428. If this is a subsection, then 4.1 and the same should be used in others. See all text.

Add future studies to conclusion section.

The manuscript could be accepted after improve.

Author Response

REVIEWER 2
Thank you for the opportunity to review.

This is a useful and original contribution to the field.

The following chapters clearly describe the course of research, suggest a new method that will be very interesting to learn the reader. The manuscript itself is well structured, figures and tables are presented and described in full.

Here I provide my remarks.

Authors must use a template. - Revisited the MDPI template and corrected where we deviated from it.

It would be better to added at the abstract: the novelty of the research, what is the gape in the knowledge your research should close, place the question addressed in a broad context and highlight the purpose of the study, summarize the articles main findings and indicate the main conclusions or interpretations. - Updated the abstract and conclusion to highlight novelty, gap, the question addressed, and the purpose of the study. The main findings are indicated more clearly now.

Abstract and conclusion must be supported by data.

Line 46. Maybe it would be better to write 1 × 104 the same way as 1 × 106? - Rewrote 10 x 10^3 to   $1 x 10^4$ (line 42), making it the same way as 1 x 10^6.

Line 91. The symbols in Equation 4 are too large. Use a template. See all text. - All equations including equation 4 (line 91) were resized to the MDPI template.

Line 147. In my opinion, the author or cite for reference 9 is missing. - Reference 9 (line 147) was reorganized in the sentence to highlight their contribution made (playing games from pixels).

Lines 347, 370, 380. What is mean “x”? Before this you used “×”. - A different variation of X was a typo for lines 347,370 and 380. It meant 'by' as the image was resized to 80 pixels tall by 80 pixels wide in size. This typo was corrected.

Line 428. If this is a subsection, then 4.1 and the same should be used in others. See all text. - Line 428 was a typo subsubsection was used instead of subsection. This was corrected.

Add future studies to conclusion section. - A future studies paragraph was added to the conclusion.

The manuscript could be accepted after improve.

RESPONSE: Thank you for your time and patience to review this paper and provide feedback. We hope the above has improved the manuscript to be accepted.

1. Revisited the MDPI template and corrected where we deviated from it.
2. Updated the abstract and conclusion to highlight novelty, gap, the question addressed, and the purpose of the study. The main findings are indicated more clearly now.
3. Rewrote 10 x 10^3 to   $1 x 10^4$ (line 42), making it the same way as 1 x 10^6.
4. All equations including equation 4 (line 91) were resized to the MDPI template.
5. Reference 9 (line 147) was reorganized in the sentence to highlight their contribution made (playing games from pixels).
6. A different variation of X was a typo for lines 347,370 and 380. It meant 'by' as the image was resized to 80 pixels tall by 80 pixels wide in size. This typo was corrected.
7. Line 428 was a typo subsubsection was used instead of subsection. This was corrected.
8. A future studies paragraph was added to the conclusion.

Reviewer 3 Report

Overall, the subject is very interesting and adequate for the journal. Some suggestions are provided for the authors' consideration.

1. The author's introduction seems to lack logic, with little discussion of research background and existing research, such as citing multiple papers without describing them.

2. Line 43, Why does the author not compare this method with other methods?

3. Figure 2 has a lower resolution and should be further improved;

4. Line 360, why choose 40 hidden neurons?

5. Please give the reasons for choosing 20 Atari games.

6. It is recommended that authors select important results/Figs for presentation in order to better express views.

7.The conclusion needs to be rearranged because it is too tedious.

Author Response

REVIEWER: Overall, the subject is very interesting and adequate for the journal. Some suggestions are provided for the authors' consideration.

REVIEWER: 1. The author's introduction seems to lack logic, with little discussion of research background and existing research, such as citing multiple papers without describing them.

RESPONSE: We revised the introduction to enhance logic, research background, and discussion of existing research.

REVIEWER: 2. Line 43, Why does the author not compare this method with other methods? - It would be interesting to compare a reduction with different methods and we want to in future studies.

RESPONSE: We had refrained from comparing this method with other approaches since extensive comparisons of various aspects of experience replay have already been conducted in prior works. Instead, we focused on the unexplored aspect of reducing experience replay capacity and visualizing experience replay samplings being used to train the agent.

REVIEWER: 3. Figure 2 has a lower resolution and should be further improved; -

RESPONSE: improved

REVIEWER: 4. Line 360, why choose 40 hidden neurons?

RESPONSE: The number of neurons was chosen by performing an informal search on games like Breakout and Space Invaders, early on we planned to use it in tasks other than Atari (thus making it large), but we decided to constrain to Atari and not any other tasks in the interest of SHAP visualization of Experience Replay.

REVIEWER: 5. Please give the reasons for choosing 20 Atari games. -

RESPONSE: (Mnih et al, 2015) chose 49 games but we were constrained by hardware so chose to follow (Fedus et al, 2020) as a minimum, However, we would be interested in comparing more games, especially with higher rule density and we plan to do so in future studies.

REVIEWER : 6. It is recommended that authors select important results/Figs for presentation in order to better express views.

RESPONSE: The results section was reconsolidated to show the important results from Dunn's test.

REVIEWER : 7. The conclusion needs to be rearranged because it is too tedious.

RESPONSE: We rearranged the conclusion for ease of reading and removed any unnecessary repetition.

Round 2

Reviewer 2 Report

Thank you for the comments. The manuscript can be accepted.